# The Effectiveness of Preoperative Outpatient and Home Rehabilitation and the Impact on the Results of Hip Arthroplasty: Introductory Report

**DOI:** 10.3390/healthcare12030327

**Published:** 2024-01-26

**Authors:** Adam Zdziechowski, Magdalena Zdziechowska, Jacek Rysz, Marta Woldańska-Okońska

**Affiliations:** 1Department of Internal Diseases, Rehabilitation and Physical Medicine, Medical University of Łódź, 90-419 Łódź, Poland; marta.woldanska-okonska@umed.lodz.pl; 2Department of Nephrology, Hypertension and Family Medicine, Medical University of Łódź, 90-419 Łódź, Poland

**Keywords:** hip arthroplasty, preoperative rehabilitation, prehabilitation, hip osteoarthritis, effectiveness of prehabilitation

## Abstract

A total hip replacement is the treatment of choice for end-stage hip osteoarthritis. Rehabilitation performed before surgery (called prehabilitation) is used to improve the results of surgical treatment. However, the results of studies have not unquestionably confirmed the effectiveness of preoperative rehabilitation and its impact on the outcome of surgery. The aim of this study is to assess the effectiveness of preoperative outpatient and home rehabilitation in relation to a control group not subject to these forms of influence. A total of 61 patients qualified for primary hip arthroplasty were randomly assigned to a group with outpatient rehabilitation before surgery, exercises performed at home, or a group without any intervention before surgery. Three weeks after surgery, the patients were re-qualified and underwent three weeks of outpatient rehabilitation in the day rehabilitation department. The patients from all three groups were evaluated in terms of functionality and pain using point scales upon enrolment in the study, on admission to the day rehabilitation department, and after 3 weeks of rehabilitation in the department. A total of 50 subjects completed the study. The study results did not reveal statistically significant differences between preoperative rehabilitation and no intervention. Patients rehabilitated at home gave up self-therapy more often than those undergoing outpatient rehabilitation.

## 1. Introduction

Osteoarthritis of the hip joint—coxarthrosis—is one of the most common causes of sickness absence among men in Poland [1] (data from 2019—before the outbreak of the SARS-CoV-2 pandemic). Osteoarthritis of the hip joint is recognized as the second most common location of degenerative changes in the joints of the limbs (it is most often diagnosed in the knee joint) and is the most common reason for total joint replacement both in Poland and worldwide [2]. Even though more than 130 years have passed since the first arthroplasty [3], the effects of this form of treatment are not satisfactory. Rehabilitation is one of the basic methods of treating early coxarthrosis [4]; physical therapy is also widely used to improve the functional status after hip arthroplasty [5].

The use of prehabilitation—rehabilitation preceding treatment—has features that make this form of influence potentially attractive: ensuring patient education [6], increasing muscle strength and range of motion [7,8], and improving proprioception [9]. The prehabilitation efficacy evaluation has major importance considering coxartrhosis prevalence. Recently, prehabilitation is becoming increasingly popular as a method that potentially supports broadly understood oncological treatment [10,11,12]. However, prehabilitation only makes sense when it significantly improves the patient’s functional status and has a positive impact on the result of the final treatment. Moreover, prehabilitation generates costs both for the health care system and the patient [13]. Medical and cost effectiveness are therefore of key importance if prehabilitation is to become a separate service included in health insurance [14]. Currently, rehabilitation before surgical treatment in Poland is not a routine procedure, although some orthopaedists, rehabilitation specialists, and physiotherapists consider it justified to recommend it to patients.

The assessment of the effectiveness of prehabilitation in patients using self-therapy, postulated already before the pandemic, became more important in 2020 [15,16]. A rather conservative form of communication—a paper brochure with an outline of exercises (see Appendix A)—has been replaced by therapy conducted via electronic media, including smartphones commonly available in developed countries. However, the paper form has the advantage of accessibility and ease of use; after all, having a smartphone and the ability to use the application fluently is not obvious, especially among the elderly. It is postulated to standardize the methodology for creating medical applications used by patients [17,18]. Certainly, applications enable their creators to automatically remind patients about the need to perform a given activity, a functionality that paper instructions do not have.

The Postel Merle D’Aubigne score (PMA) requires the assessment of the range of abduction and flexion of the hip joints, i.e., the critical range of motion for efficient movement up stairs and restoring an energy-efficient gait stereotype without excessive, harmful movement of the center of gravity in the frontal plane [19]. It is a reliable, repetitive tool in clinical practice [20]. The HOOS Hip Disability and Osteoarthritis Outcome Score (HOOS) score allows for a comprehensive assessment of the patient’s condition, from pain to sports activities. The Polish version of HOOS has been positively evaluated for use in evaluating the results of hip arthroplasty [21]. The Laitinen scale is applied to assess the severity of pain in the context of intensity, frequency, need to use analgesics and impact on physical activity. A visual analogue scale (VAS) is used to assess the subjective intensity of pain. The variant of 100 mm was chosen, where 0 means no pain at all, and 100 mm—the strongest pain experienced or imaginable to the patient.

The tools are based on the patient’s self-evaluation and functional status. The study group consisted mostly of elderly patients, for whom independent movement is one of the key conditions for maintaining independence in basic areas of functioning and playing social roles. Scales based on self-evaluation also allow for the assessment of the patient’s risk of kinesiophobia (fear of physical activity), which in the geriatric population leads to a secondary disability and reduced quality of life [22,23]. 

Given the availability of rehabilitation services and the simplicity of exercises used in self-therapy, the study was designed to compare the effectiveness of exercises performed independently with outpatient rehabilitation requiring the participation of a physiotherapist and the use of specialist equipment. As the study was prepared in order to check the efficacy of prehabilitation in specific Polish healthcare conditions, both outpatient prehabilitation and prehabilitation at home took 15 exercise days (3 weeks with a weekend break without rehabilitation). Apart from the national health fund regulation, there was also an organizational reason for establishing a 3-week rehabilitation period—the patients are usually not informed about an arthroplasty date with a long-term advance. Comparing with previous hip prehabilitation studies, the prehabilitation programs usually take longer (from 3 up to 12 weeks); however, there are few “rehabilitation days” (usually 2–3 rehabilitation sessions weekly) [24]. The Cochrane Collaboration reviews confirm exercise efficacy in hip arthrosis yet fail to establish optimal training capacity [25,26]. The intensity and complexity of physiotherapy are a sufficient difference compared to the self-therapy scheme. Osteoarthritis of the hip joint in most patients causes similar deviations and limitations in the range of motion of the hip joint and, consequently, the entire musculoskeletal system [27,28]. The use of local cryotherapy is characterized by high safety and good tolerance [29]. Contraindications are quite rare, especially in patients qualified for major elective surgery—hip arthroplasty [30]. Post-isometric relaxation (PIR) of some muscles requires the use of an exercise table and instructions from the physiotherapist [31]. PIR is applied to increase the length of excessively shortened muscles in order to improve muscle balance and the work of antagonistic muscles [32]. This kind of physiotherapy can be performed by any qualified physiotherapist without the need for additional training. The use of commonly known and available physiotherapy procedures was intended to reveal the differences between easy-to-perform self-therapy and demanding professional-staff outpatient rehabilitation.

As a rule, rehabilitation after hip arthroplasty should begin in the orthopaedics department. Early postoperative rehabilitation shortens the patient’s stay in the surgical ward and does not increase the risk of complications [33]. Physiotherapy conducted in the orthopaedics departments brings tangible benefits compared to patient education alone [34].

## 2. Materials and Methods

The study included 61 patients with osteoarthritis of the hip joint qualified for elective total hip replacement, which allowed for the introduction of preoperative rehabilitation. The study was conducted from June 2018 to December 2022 in the day rehabilitation department of the Intermedicus Rehabilitation Centre “Janosik”. The inclusion and exclusion criteria for the study are presented in Table 1. The exclusion criteria largely coincide with the contraindications to hip arthroplasty. 

After enrolling in the study and giving consent, the patients were randomly assigned to three groups. The research randomizer tool (available on the website https://www.randomizer.org/) was used accessed on 6 July 2018. Afterwards, the subject’s health status was assessed clinically and using point scales (visual analog scale VAS, Laitinen scale, HOOS scale, PMA score). The treatment effect in this study was assessed using the Merle d’Aubigné and Postel (PMA) score, the HOOS score, the Laitinen scale, and the visual analog scale (VAS). The six-minute walk test was rejected because it was impossible to conduct the examination in accordance with its methodology due to the space limitations of the facility where the study was conducted. The HOOS questionnaires and the VAS scale were completed in the presence of the researcher by the patient, while the Laitinen scale and PMA were applied by the researcher. The hip range of motion (ROM) was measured by the researcher using a goniometer. The ROM assessment may reveal the inconsistency between joint function improvement and patients’ outcome assessments. Both VAS and HOOS are patient-reported tools that are functionally compliant with the patient-reported outcome measures. (PROMs) were promoted, albeit introduced before PROMs were defined. Whereas HOOS is dedicated exclusively to hip dysfunction evaluation, VAS is an intuitive and universal tool used in different diseases and contexts [35,36,37]. A calculator available on the website (available at https://orthotoolkit.com, accessed on 6 July 2018) was used to calculate the result obtained using the HOOS scale. 

Group I consisted of patients who, before the elective surgery, were treated in a day rehabilitation unit according to an individual schedule lasting for 15 working days (with breaks on Saturday and Sunday). Group I underwent the following forms of physiotherapy: cryotherapy with liquid nitrogen of the hip joint intended for surgery (before kinesiotherapy, the duration of the procedure was 2–3 min), active weight-bearing exercises with graduated resistance to the hip joints in the frontal and sagittal plane (about 15 min), mobilization and traction of the hip joint planned for surgery (5–10 min), post-isometric relaxation of the iliopsoas muscles, rectus femoris muscles, external rotators of the hip joints (total about 10–20 min), obstacle walking training, and learning to walk with elbow crutches (approximately 15 min). The treatment program used procedures that would not be feasible at home but could be implemented in virtually any facility providing physiotherapy services under a contract with the National Health Fund.

Group II consisted of patients who received a paper exercise outline and one-time instructions to practice on their own at home. Group II patients independently performed: active exercises of the hip joints in isolated positions; exercises to relax the external rotator muscles of the hip joint; exercises to relax the iliopsoas and rectus femoris muscles; and isometric exercises of the abdominal and buttocks muscles. The patient was to exercise 5 days a week, for a total of 15 days. In order to ensure reliable performance of the exercises, the patients were asked to perform them collectively once a day, with the suggested exercise up to the point of pain for 30–40 min.

Group III consisted of patients who had not undergone any rehabilitation intervention before elective arthroplasty, apart from the initial assessment of the health condition using the same tools as in the other groups.

At the time of recruitment to the study, all patients were scheduled for surgery approximately 26–34 days after qualification for the study.

Patients from all three groups were subjected to rehabilitation in the day rehabilitation unit 13–16 days after surgery. According to the Polish Health Technology Assessment and Tariffication Agency, the first control visit after discharge with wound control and removing surgical sutures should be conducted 12–14 days after discharge [38]. In the time between hospital discharge and admission to an outpatient rehabilitation center, the patients remained active and performed the exercise program routinely explained and introduced after an alloplasty in the hospital. The detailed recommendations by the surgeon might be different depending on the patient’s health status [38]. During the first day of stay in the day rehabilitation department, the patient’s health condition was reassessed. After 14–16 days of post-operative rehabilitation, the patient’s health was assessed again. Postoperative physiotherapy was conducted based on the medical history, the results of physical examination and the patient’s preferences, with 5–6 physiotherapy procedures a day.

From the moment of qualification for the study, the subjects from all groups had an opportunity to undergo a medical consultation.

A follow-up study with an evaluation of the 3 years after the surgery period is being prepared.

## 3. Results

### 3.1. Data Analysis

The results were presented as mean ± standard deviation (mean ± SD) for continuous variables with a normal distribution or as median with quartile range (median, 25–75%) otherwise. Categorical variables were shown as percentages in relation to the size of the study group. The percentage change in individual parameters during therapy was calculated based on the following formula: (T_1_ − T_0_) × (T_0_ × 10^−1^) × 100% 

T_0_—value before intervention

T_1_—value after intervention

The normality of the distribution of variables was verified based on the Shapiro-Wilk test. In the case of a normal distribution with no difference in variances, ANOVA followed by Tukey’s post-hoc multiple comparison tests were applied to assess the significance of differences between the three groups, and for data deviating from a normal distribution, the Kruskal-Wallis test and the Conover-Iman post-hoc test were used. The significance of differences for paired data (different time points) was analyzed using the Wilcoxon test. For statistical analysis of non-continuous results, the chi-square test of independence, or Fisher’s exact test, was applied.

### 3.2. Results

A total of 50 people completed the study. All patients were operated on via a posterolateral approach. Of the 11 people who did not complete the study, 6 were from the home rehabilitation group. Five people from the home rehabilitation group claimed that they were unable to meet the requirements of the study because the training program was too intense. One person from the home rehabilitation group had a prolonged postoperative stay, ultimately having a revision of the operated joint and revision arthroplasty. Three people from the preoperative rehabilitation group terminated their participation in the study prematurely. One person from the prehabilitation group resigned for random reasons, and two were afraid of a COVID-19 infection in the rehabilitation center. Two people from the group without preoperative rehabilitation resigned from participation (one person was afraid of COVID-19 infection in the rehabilitation center; one resigned from arthroplasty). None of the people who completed the study experienced any side effects from the rehabilitation. The patients expressed positive opinions about both pre- and postoperative rehabilitation, although the assessment of these opinions was not the purpose of the study. The patients from the home rehabilitation group who succeeded in completing the exercise program denied serious subjective improvement after finishing prehabilitation. On the other hand, these patients found the exercise did not cause fatigue, which may be due to the small exercise capacity and intensity. The collected data are presented in Table 2, Table 3, Table 4 and Table 5.

All three study groups did not differ significantly from each other, and this was confirmed by the ANOVA or Kruskall—Wallis test (Table 5). The patients from all 3 groups benefited from the surgical treatment, however, no statistically significant differences were found between the groups. This observation comes from the ANOVA analysis (percentage of improvement compared to the baseline value), which did not reveal statistically significant differences in any of the parameters. The majority of patients was recruited after COVID-19 onset. The additional statistical analysis revealed no significant differences in treatment results or group characteristic between patients recruited before and after the pandemia onset. 

## 4. Discussion

The results of the study are partially consistent with the ambiguous assessment of prehabilitation shown in previous studies [39,40]. There were almost no differences between the study groups in any aspect. The lack of significant differences between the samples is an argument against the use of prehabilitation in the adopted model. Apart from statistical and methodological reasons, the ineffectiveness of prehabilitation may have other causes.

Rehabilitation (and prehabilitation as a modified form of classic rehabilitation) balances between attempts to standardize treatment in specific clinical situations and striving for individualization and optimal adjustment of the therapy for a given patient [41]. There are common features typical of a patient with advanced coxarthrosis (weakened muscle strength, limited range of motion in the hip joint) [42]. The severity of the symptoms varies significantly between individuals due to environmental and genetic factors [43,44]. Most patients with osteoarthritis of the hip qualified for arthroplasty are elderly or middle-aged people [45]. They differ in co-morbidities, broadly understood physical fitness, lifestyle, and preferences. Also important are the biomechanical features established over the years, such as the shape of spine curvatures and the properties of soft tissues, which are difficult to evaluate clinically [46]. The psychosocial diversity of patients should also be taken into account [47]. Self-esteem, motivation (or the lack thereof) for a complete return to activity, and job satisfaction influence the effects of therapy, which were confirmed by the results of previous studies [48,49,50]. The patient perspective focuses on meeting expectations (that differ among individuals) and pain relief. That leads to inconsistency between patient satisfaction and observed functional improvement [51,52]. Given the above variables, individualization of training in order to obtain the maximum therapeutic effect seems to be a necessity. The individualized approach demands further investigation. However, an adequate study design for individualized therapy is a challenge for methodological reasons [53].

Most patients qualified for hip arthroplasty experience significant intensification of the symptoms of the disease, which negatively affects the potential for functional improvement, muscle strengthening, and correcting the gait pattern [54,55]. The initial evaluation of patients from all three groups qualified for the study confirms this thesis. The negligible potential of positive changes achieved in the advanced stage of the disease seems to be consistent with the results. Attempts to improve treatment results by means of prehabilitation shortly before surgery may even intensify the symptoms and cause suffering in some patients. Perhaps therapeutic education should become the basic form of treatment in the short period before hip arthroplasty [56,57]. The patient’s decision aids may improve the patient’s educational status and influence his or her expectations [58]. Patient decision aids are not commonly used in Poland for hip alloplasty qualification.

Home rehabilitation offers serious advantages like a convenient schedule, an increase in patient confidence, and intimacy [59]. The use of a paper leaflet generates low costs, however, which does not support controlling the rehabilitation process available while using “digital rehabilitation” [60]. Home telerehabilitation applications require a change in the attitude of both patients and health professionals that exceeds the issue of high-tech usage ability [61,62]. Some studies suggest that healthcare is not yet ready for widespread telerehabilitation, as both physiotherapy students and educators lack the competence to gain whole-person telerehabilitation potential [63,64]. The authors agree that home telerehabilitation will develop rapidly, replace paper leaflets or popular internet services, and improve healthcare quality worldwide.

The time of the study coincided with the COVID-19 pandemic. A total of 41 patients participated in the study during the pandemic (from 20 March 2020 to 16 May 2022) or during the state of epidemiological threat (from 16 May 2022, the state of epidemiological threat was lifted on 1 July 2023). The pandemic has drastically impacted the functioning of the medical care sector around the world, including the procedures of rehabilitation and elective surgeries [65]. Many studies indicated the insufficient ability of departments of orthopaedics to perform elective arthroplasty, significantly reduced during the pandemic, thus considerably diminishing the quality of life of patients forced to wait longer for the procedure. During the pandemic, there was an increased interest in forms of treatment that did not require patients to visit health care facilities, including virtual reality and remote image transmission [66,67]. Some patients were vaccinated before joining the study group. The majority of patients who developed severe COVID complications were unable to undergo hip arthroplasty for safety reasons. Thus, we consider that COVID decreased the number of patients recruited to the study but did not have a big impact on the effects observed in the study. Certainly, artificial intelligence and software created by companies specializing in medical applications will irreversibly revolutionize the rehabilitation of hip arthroplasty in the future [68,69].

### Limitations of the Study

The study was conducted on a small sample. The results will be used for power analysis for further study. In the Polish health care system, it is difficult to plan prehabilitation and rehabilitation after surgery within a strict time window for organizational reasons. The healthcare regulations do not promote postsurgical rehabilitation as a priority in queuing, which limits access to rehabilitation services for recently operated patients.

Most of the study was conducted during the COVID-19 pandemic, which probably reduced the number of recruited patients. A more long-term evaluation of treatment outcomes would provide an opportunity to more fully assess the effects of the intervention. Differences between the groups may become visible in statistical studies with a longer observation period. The study was conducted in one center because multi-center research was impossible due to a lack of funding sources and interest from institutions not conducting research and teaching activities. The patients had operations performed on different orthopaedic wards by different surgical teams. As home prehabilitation was not supported by telerehabilitation software and other devices, the information provided by the patients could not be verified. In future studies, digital devices should ensure information concerning patient engagement and efforts.

## 5. Conclusions

Prehabilitation, according to the adopted model, should not be recommended as it does not improve the results of surgical treatment. Individualization of the training program may increase the effectiveness of prehabilitation. Prehabilitation, which is a form of treatment based on physiotherapy applied shortly before surgery, has no effect on the patients’ condition, which is probably due to the limited possibilities of improving the health condition because of the severity of the symptoms of the disease.

## Figures and Tables

**Table 1 healthcare-12-00327-t001:** Inclusion and exclusion criteria for the study.

Inclusion Criteria for the Study	Exclusion Criteria from the Study
Primary osteoarthritis of the hip joint	Pregnancy
Qualification for total hip arthroplasty	Previous arthroplasty of any lower limb joint
Known date of the procedure	Osteoporosis
Patient’s ability to reach the day rehabilitation unit	Unstable ischemic heart disease
Informed consent to participate in the study	Unstable heart failure
	Arterial hypertension not controlled pharmacologically
	Lung diseases that limit gas exchange
	Active infectious diseases
	Malignant tumors diagnosed during treatment
	Cancers whose treatment was completed within one year of recruitment to the study
	A history of thromboembolic event within the last 6 months
	Renal failure requiring dialysis
	Glaucoma qualified for surgical treatment
	Addictions that make it difficult to perform social functions
	Progressive neurological diseases (including multiple sclerosis, amyotrophic lateral sclerosis, Parkinson’s disease)
	Significant limitation of motor coordination
	Paresis due to damage to the central and/or peripheral nervous system
	Significantly advanced systemic diseases (systemic lupus erythematosus, collagen diseases, rheumatoid arthritis, psoriatic arthritis and others)
	Other diseases of unknown etiology

**Table 2 healthcare-12-00327-t002:** The results of patients receiving no intervention before surgery.

No Prehabilitation N = 20 (Group 3)	Before Surgery	After Surgery	After Surgery and Rehabilitation	Significance
PMA—range of motion	3.0 (2.0–3.0)	3.5 (3.0–5.0) **	5.0 (4.0–5.7) ##	*p* < 0.0001
PMA—pain intensity	2.0 (1.0–2.7)	3.0 (3.0–4.0) **	5.0 (4.2–6.0) ###	*p* < 0.0001
Laitinen scale	11.0 (10.2–12.0)	8.5 (7.2–10.7) ***	3.5 (1.2–6.0) ###	*p* < 0.0001
VAS—visual analogue scale	78.5 (71.0–85.0)	68.0 (55.0–78.7) **	26.5 (15.0–45.0) ###	*p* < 0.0001
HOOS—symptoms, stiffness	25.0 (15.0–30.0)	52.5 (36.2–73.7) ***	80.0 (66.2–90.0) ###	*p* < 0.0001
HOOS—pain	20.0 (10.6–25.0)	30.0 (27.5–45.0) **	73.7 (50.6–85.0) ###	*p* < 0.0001
HOOS—everyday activity	19.1 (13.6–22.1)	33.1 (22.1–46.0) **	77.2 (47.1–83.8) ###	*p* < 0.0001
HOOS—sports activity	9.4 (2.3–23.4)	18.8 (12.5–37.5) **	37.5 (31.3–67.2) ##	*p* < 0.0001
HOOS—quality of life	12.5 (6.3–18.8)	31.3 (18.8–43.8) ***	65.6 (50.0–79.7) ###	*p* < 0.0001
HOOS—total score	19.7 (14.1–22.9)	35.0 (22.5–45.3) **	71.9 (49.2–80.7) ###	*p* = 0.001

The results obtained in the group of patients without any intervention before surgery, presented as median with a quartile range of 25–75%. The significance given as * refers to the comparison of postoperative results before postoperative rehabilitation and preoperative results (** *p* < 0.01, *** *p* < 0.0001). The significance presented as # refers to the comparison of results after surgery with those after postoperative rehabilitation (## *p* < 0.01, ### *p* < 0.0001). Statistical significance was estimated by the Wilcoxon test.

**Table 3 healthcare-12-00327-t003:** The results of patients training at home.

Rehabilitation at Home N = 9 (Group 1)	Before Surgery	After Surgery	After Surgery and Rehabilitation	Significance
PMA—range of motion	2.0 (2.0–2.0)	4.0 (2.5–4.0) *	5.0 (4.0–5.0) ##	*p* = 0.007
PMA—pain intensity	2.0 (1.5–3.0)	4.0 (3.0–5.5) *	6.0 (4.0–6.0) #	*p* = 0.007
Laitinen scale	10.0 (8.0–12.5)	8.0 (5.5–10.0) **	3.0 (0–4.0) #	*p* = 0.008
VAS—visual analogue scale	75.0 (64.0–84.0)	66.0 (53.5–71.0) **	28.0 (14.0–31.5) ##	*p* = 0.008
HOOS—symptoms, stiffness	25.0 (15.0–42.5)	65.0 (37.5–85.0) *	80.0 (57.5–87.5) #	*p* = 0.008
HOOS—pain	22.5 (12.5–33.7)	32.5 (27.5–53.7) NS	67.5 (62.5–77.5) ##	*p* = 0.008
HOOS—everyday activity	17.6 (15.4–30.1)	33.8 (26.5–41.9) *	63.2 (55.1–75.0)	*p* = 0.008
HOOS—sports activity	12.5 (6.3–25.0)	25.0 (15.6–34.4) *	62.5 (46.9–68.8) ##	*p* = 0.008
HOOS—quality of life	12.5 (6.3–25.0)	37.5 (25.0–50.0) *	68.8 (62.5–78.1) ##	*p* = 0.007
HOOS—total score	19.4 (17.8–27.5)	38.1 (32.8–41.2) *	65.0 (55.9–74.0)	*p* = 0.008

The results obtained in the group of patients with home rehabilitation before surgery were presented as median with a quartile range of 25–75%. The significance presented as * refers to the comparison of the results after surgery before postoperative rehabilitation, and the results before surgery (NS-statistically insignificant; * *p* < 0.05; ** *p* < 0.01). The significance presented as # refers to the comparison of the results after surgery with the results after postoperative rehabilitation (NS-statistically insignificant; # *p* < 0.05, ## *p* < 0.01). Statistical significance measured by the Wilcoxon test.

**Table 4 healthcare-12-00327-t004:** The results of patients receiving outpatient rehabilitation before surgery.

With Rehabilitation before Surgery N = 21 (Group 2)	Before Surgery	After Surgery	After Surgery and Rehabilitation	Significance
PMA—range of motion	2.0 (1.5–3.0)	4.0 (3.0–4.0) ***	5.0 (4.0–5.0) ##	*p* < 0.0001
PMA—pain intensity	2.0 (1.0–2.0)	3.0 (2.0–4.0) **	5.0 (4.0–6.0) ###	*p* < 0.001
Laitinen scale	11.0 (10.0–13.0)	9.0 (8.0–12.0) **	4.0 (1.5–8.0) ###	*p* < 0.001
VAS—visual analogue scale	75.0 (65.5–88.0)	70.0 (60.0–77.5) **	30.0 (13.5–52.5) ###	*p* < 0.001
HOOS—symptoms, stiffness	25.0 (15.0–35.0)	50.0 (35.0–65.0) ***	90.0 (77.5–95.0) ###	*p* < 0.0001
HOOS—pain	25.0 (22.5–30.0)	40.0 (27.5–55.0) ***	87.5 (68.7–97.5) ###	*p* < 0.0001
HOOS—ADL	27.9 (23.5–32.3)	38.2 (30.1–47.0) ***	86.8 (55.9–91.9) ###	*p* < 0.0001
HOOS—sports activity	12.5 (6.3–18.8)	31.3 (18.8–37.5) ***	62.5 (37.5–81.3) ###	*p* < 0.0001
HOOS—quality of life	12.5 (6.3–18.8)	37.5 (21.9–50.0) ***	75.0 (50.0–90.6) ###	*p* < 0.0001
HOOS—total score	23.1 (20.9–28.8)	37.5 (29.7–52.5) ***	84.4 (61.5–89.7) ###	*p* < 0.0001

The results obtained in the group of patients with outpatient prehabilitation before surgery were presented as median with a quartile range of 25–75%. The significance presented as * refers to the comparison of the results after surgery before postoperative rehabilitation and the results before surgery (** *p* < 0.01; *** *p* < 0.0001). The significance presented as # refers to the comparison of the results after surgery with the results after postoperative rehabilitation (## *p* < 0.01, ### *p* < 0.0001). Statistical significance is measured significance measured by the Wilcoxon test.

**Table 5 healthcare-12-00327-t005:** The comparison of the intergroup results and statistical significance of the intergroup differences.

	Without Prehabilitation N = 20	Rehabilitation at Home N = 9	With Rehabilitation before Surgery N = 21	Significance
Age	68.3 ± 9.4	69.0 ± 8.4	64.1 ± 10.5	NS (*p* = 0.290)
Gender [M]	6 (30.0%)	2 (22.2%)	9 (42.9%)	NS (*p* = 0.488)
PMA ROM (after rehabilitation vs. qualification for the study)	83.3 (50.0–100.0)	150.0 (83.3–150.0)	100.0 (50.0–250.0)	NS (*p* = 0.187)
PMA ROM (after rehabilitation vs. after surgery)	10.0 (0–66.7)	33.3 (25.0–58.3)	25.0 (0–66.7)	NS (*p* = 0.523)
MA Pain (after rehabilitation vs. qualification for the study)	200.0 (100.0–400.0)	200.0 (83.3–200.0)	200.0 (100.0–400.0)	NS (*p* = 0.736)
PMA Pain (after rehabilitation vs. after surgery)	50.0 (25.0–91.7)	20.0 (0–58.3)	50.0 (22.5–125.0)	NS (*p* = 0.242)
Laitinen scale (after rehabilitation vs. qualification for the study)	−66.7 (−87.9–41.4)	−72.7 (−100.0–59.0)	−60.0 (−85.1–25.2)	NS (*p* = 0.329)
Laitinen scale (after rehabilitation vs. after surgery)	−52.8 (−70.2–27.1)	−63.6 (−85.0–31.4)	−44.4 (−70.2–19.1)	NS (*p* = 0.540)
VAS (after rehabilitation vs. qualification for the study)	−69.0 (−84.2–37.9)	−62.9 (−81.8–58.6)	−56.9 (−81.2–32.0)	NS (*p* = 0.739)
VAS (after rehabilitation vs. after surgery)	−54.2 (−77.8–26.5)	−57.6 (−80.2–31.4)	−49.3 (−76.4–15.1)	NS (*p* = 0.776)
HOOS—symptoms, stiffness (after rehabilitation vs. qualification for the study)	218.3 (121.9–418.7)	220.0 (85.0–391.6)	220.0 (178.5–391.6)	NS (*p* = 0.680)
HOOS—symptoms, stiffness (after rehabilitation vs. after surgery)	30.0 (10.6–108.3)	23.1 (2.9–60.3)	60.0 (24.3–119.8)	NS (*p* = 0.150)
HOOS—pain (after rehabilitation vs. qualification for the study)	255.8 (157.8–501.2)	237.5 (100.6–390.0)	218.2 (151.4–281.2)	NS (*p* = 0.412)
HOOS—pain (after rehabilitation vs. after surgery)	81.1 (46.6–136.3)	107.7 (18.8–141.5)	81.3 (46.7–135.2)	NS (*p* = 0.956)
HOOS—ADL (after rehabilitation vs. qualification for the study)	274.4 (149.5–363.6)	277.5 (76.3–340.5)	169.8 (130.6–248.0)	NS (*p* = 0.104)
HOOS—ADL (after rehabilitation vs. after surgery)	61.2 (32.8–175.7)	94.3 (47.8–130.3)	77.0 (50.6–182.1)	NS (*p* = 0.779)
HOOS—sports activities (after rehabilitation vs. qualification for the study)	347.8 (118.6–1085.5)	250.4 (125.8–942.1)	332.4 (189.8–622.0)	NS (*p* = 0.980)
HOOS—sports activities (after rehabilitation vs. after surgery)	66.6 (2.2–289.1)	99.7 (36.6–283.0)	75.0 (37.8–126.5)	NS (*p* = 0.655)
HOOS—Qol (after rehabilitation vs. qualification for the study)	407.9 (212.6–991.3)	450.4 (177.4–793.7)	500.0 (297.8–746.8)	NS (*p* = 0.890)
HOOS—Qol (after rehabilitation vs. after surgery)	77.3 (19.8–166.3)	116.8 (49.2–187.6)	75.0 (16.4–199.7)	NS (*p* = 0.699)
HOOS—total (after rehabilitation vs. qualification for the study)	258.7 (179.0–386.6)	219.1 (114.7–308.9)	218.4 (179.6–279.5)	NS (*p* = 0.429)
Total (after rehabilitation vs. after surgery)	70.1 (30.4–173.0)	90.4 (45.4–94.1)	84.7 (51.4–133.1)	NS (*p* = 0.715)

The analysis of intergroup differences. The results are presented as a percentage change compared to the initial examination performed during recruitment to the study. Statistical significance is measured by ANOVA or Kruskal–Wallis test (NS—statistically insignificant).

## Data Availability

Data are unavailable due to privacy or ethical restrictions.

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
