# Peer review of "The Effectiveness of Preoperative Outpatient and Home Rehabilitation and the Impact on the Results of Hip Arthroplasty: Introductory Report"

_healthcare, 2024, doi:10.3390/healthcare12030327_

Round 1

Reviewer 1 Report

Comments and Suggestions for Authors

Dear authors, thank you for the opportunity to give this review.

The importance of rehabilitation programs before orthopedic surgery is very well reflected in the paper. The introduction provides sufficient data on the aim of the study, regarding rehabilitation programs in hip surgery, especially arthroplasty.

The methods are very clearly presented, but the final number of subjects included in the study is low.

Also, I didn't find any comments or special plans/methods for monitoring the batch with home rehabilitation programs. It was telerehabilitation? Or using other devices?

Results are presented only in tables, no graphics. Can be added?

The discussion and conclusions are well presented.

Author Response

Dear Reviewer,

thank You for well-considered suggestions.

We were unable to obtain bigger population to our study. The COVID pandemia and organizational lacks (patient get the information about operation date far too late). The research was performed in one rehabilitation center and, for formal reasons, it was not possible to add other outpatient clinic to the research. The authors agree, the bigger sample may give different results in statistical analysis.

Unfortunately, the telerehabilitation was not accessible for the study purposes and the paper leaflet was used. The lack of reliable control of the home based rehabilitation was added in study limitations section. It is emphasized in revised article version.

The leaflet for the in English version is provided in the attachment.

Of course, the graphical data can be added, but we tried to avoid duplicating the data in few places of study paper.

Reviewer 2 Report

Comments and Suggestions for Authors

Dear authors,

a fair try for an interesting topic but language editing is mandatory. Moreover, regarding the methodology, COVID-19 pandemic influence is really high to this orthopaedic study. As far as I am concerned, this period should be excluded.  Introduction section is way too long, please reform it to a more focus version.  On contrary, discussion section needs further expanding. 

Comments on the Quality of English Language

Dear Authors,

Quality of english negatively affects comprehension of the text. There are many points where is difficult to understand the meaning of the text.

Line 8: advanced is possible severe

Lines 12-13: Control group ?

Line 28: Sickness absence among men ? Return to work time? Only male patients?

Lines 37-39: Unclear meaning

Lines 44-45: Unclear meaning

And so on

Author Response

Dear Reviewer,

Thank You for well-reasoned suggestions that help us improving the article

The COVID influence was crucial for all medical services. However, the exclusion of COVID period will result in much smaller statistical accuracy of data collected. We can try to divide the patients into COVID period and non COVID period and compare the results, thus such approach may also cause further doubts.

The COVID period was varied – we observed the major difficulties in providing rehabilitation in 2020. The outpatient rehabilitation was closed in march till may of 2000. The rehabilitation hospital departments were transformed into COVID wards. Planned hip arthroplasties (and other planned operations) were often delayed due to COVID occurrence in orthopedic wards. In the beginning of 2021 started COVID vaccination program and some patients became more eager to participate in rehabilitation and have arthroplasty performed.

The patients who completed participation in whole study did not develop COVID during study. Some patients suffered from COVID before participation in the study and developed mild, transient symptoms. Some patients were vaccinated before joining the study group. The majority of patients who developed severe COVID complications was unable to hip arthroplasty for safety reasons. Thus, we consider the COVID decreased the number of patients recruited to the study but did not have big impact on effects observed in the study.

The discussion section is expanded.

The article was checked for language correction by translator specialized in medical language. The text will be reexamined after consideration suggestions of Editors and edition of the article.

Reviewer 3 Report

Comments and Suggestions for Authors

a. introduction does not clearly establish the link between the prevalence of hip osteoarthritis and the specific focus of the study, which is the effectiveness of prehabilitation.

b. There is a lack of clarity on why prehabilitation is being compared to self-therapy and outpatient rehabilitation

c. section discussing the introduction of ACHE (lines 70-77) seems vague unless its relevance to the study focuses on hip arthroplasty and prehabilitation is more clarified

d. The authors study design does not specify how the random assignment of patients to the three groups was carried out

e. There is no mention of how the adherence to the exercise regimen in Group II (home-based exercises) was monitored or verified

f. There is no mention of a follow-up period to assess long-term outcomes in the methods section

g. row 250 - The argument against prehabilitation based on the lack of significant differences is not fully substantiated with specific data or analysis

Author Response

Dear Reviewer,

thank you very much for your insightful and valuable comments, which are relevant to the presented article.

The prevalence of disease and prehabilitation will be emphasized.

The background of the comparison will be explained.

The ACHE section will be removed from the text. It shows different approach to treatment optimalisation but in fact does not refer to the article. The introduction section is presented in more compact and strict way after revision.

The randomization procedure was briefly explained in lines 141-142 of the edited version.

The exercises were presented to the patient during the visit after assignment to group 2. The leaflet material (translated form is in the attachment) was a reminder given to the patient. The study was to compare the effects of training led by the physiotherapist to the exercise performed at home after presenting the training program but without supervision.  As the home rehabilitation was evaluated basing on subjects’ truthfulness, the lack of appropriate control tool decrease the study value. It is mentioned in edited version.

The long - term period assessment is possible as the authors have access to patients’ phone numbers. It is possible to perform questionnaire by phone, whereas physical examination demands patients’ attendance. Many patients finish the treatment after arthroplasty even though their quality of life and fitness demands intervention. The 3 year follow up study is prepared.

The data statistical analysis revealed no intergroup differences. Probably bigger sample and additional parameters may be helpful in improving the overall quality of research. It is clarified in edited version of the article.

Reviewer 4 Report

Comments and Suggestions for Authors

This article delves into the impact of preoperative outpatient and home rehabilitation on hip arthroplasty outcomes. The conclusion drawn is that neither preoperative outpatient rehabilitation nor home-based exercise rehabilitation before a hip arthroplasty operation yields positive outcomes in terms of patient functionality and pain. However, the article occasionally introduces extraneous details, lacks focus on key points, and may confuse readers. Therefore, significant corrections are needed.

1. References: Half of the introduction sentences are missing references. It’s almost impossible to review a paper without references. How can I double-check your statements?
- L35-36: Safety of what? Add reference.
- L37-40: Add reference.
- L41-43: Please add a reference, and I stop here asking for the reference. Please add all references missing throughout the whole text.

 2. Introduction Length: Please shorten the introduction. Too much unnecessary information. I would suggest removing unnecessary sentences that don’t have references.

3. L53: Unfortunately, I don’t have access to the appendix. 

4. L78-81: That should be included in the Methods section and not in the introduction. 

5. L194-195: Please remove this sentence to avoid potential misunderstanding by readers who may skim the text, focusing only on key phrases.

6. L198-207: Please be more precise with the numbers and clarify better which patient belongs to which group. 

7. Tables 2, 3, and 4: Please explain abbreviations and add units of measure. 

8. Regarding Group II: Have you received any feedback regarding the compliance of the participants to the home-based exercise? If yes, please explain. 

9. Do you have ethical approval for performing this project? If yes, please provide it through the manuscript

Comments on the Quality of English Language

Moderate editing of English is required. For example:

L27: cause[s]
L122-126: clarify the structure please
L198: rephrase the first sentence please
L211: table[s]

Author Response

Dear Reviewer,

thank you very much for your insightful and valuable comments, which are relevant to the presented article.

Ad. 1.2 The references will be added, the introduction will be more strict.

Ad.3 The appendix in English version will be added.

Ad.4 The lines are moved to methods section.

Ad.5 The sentence is removed.

Ad.6 The section was improved.

Ad.7 The abbreviation and units are explained.

Ad. 8 The compliance of the exercise will be discussed. We did not use any specific tool for compliance assessment but we asked patients for information concerning feasibility of program and their opinions. As the home rehabilitation was evaluated basing on subjects’ truthfulness, the lack of appropriate control tool decrease the study value. We had no access to telerehabilitation tools for monitoring purposes. It is mentioned in the edited version.

Ad.9 The Bioethics Committee approval has been mentioned in Institutional Review Board Statement lines 349-351 of the text (revised version).

The language correction will be performed in order to remove all the inaccuracies.

Reviewer 5 Report

Comments and Suggestions for Authors

Review manuscript: 2773832

The presented research article by Zdziechowski et al., reports the effectiveness of prehabilitation and its impact on the outcome of a total hip replacement surgery. Three intervention groups such as No prehabilitation, prehabilitation at home and prehabilitation at rehabilitation centre were evaluated based on their prehabilitation procedures. Despite the topic of interest, the present study does not meet the scope of the current journal requirement as it mainly lacks novelty in its study methods and the results were not intriguing. Further, the present study results were based on a small population, in addition to the limitations mentioned by the authors, making it unsuitable to consider the manuscript in the current format. Additionally, some formatting and language issues were observed.

Below are some of my specific comments, which may help the authors to improve the manuscript:

  1. The present study outcomes were obtained from a small population (50 subjects), and I think a larger sample size may be needed to capture the effectiveness of prehabilitation methods more accurately. Indeed, making conclusions using a small dataset might not be significant for this kind of assessment as many of the previous studies already reported such controversial results. Consider using a large population for better conclusive results.

  1. I believe that most of the prehabilitation programs are a kind of generalized protocol, however individual patient responses and needs can vary widely and thus it can affect the overall study results. Therefore, have authors considered/tried providing personalized approaches to effectively estimate the significance of the prehabilitation procedures based on the patient’s requirements? Such studies might be helpful for better characterization of prehabilitation procedures to improve patient's health. 

  1. Similarly, the impact of prehabilitation on surgical outcomes can vary among individuals. While it may contribute to improved outcomes in some cases, its effectiveness may be less pronounced in others, depending on factors such as fitness level and overall health. Could authors comment on such factors that could give us the variability and corresponding impact on the present study outcomes?

  1. Table 2 significance values look the same. 

  1. Could authors explain the reasons for the ineffectiveness of prehabilitation apart from the methodological reasons?

  1. It looks like the authors have used both pre- and rehabilitation interchangeably. Sometimes it is a bit confusing and try to use prehabilitation if the authors want to address pre-operative rehabilitation procedures. To be clear, change the groups to “Without prehabilitation”; “prehabilitation at home”; “prehabilitation at Centre” for consistency.

Other comments: 

1.     No Appendix was attached. Try to outline the exercise protocol.

2.     Line 70: more increasingly, 

3.     Line 79: HOOS and VAS were not defined.

4.     Line 150: walking,

5.     Line 183: Tukey’s 

6.     Line 184: Tests

7.   Line 237; The sentence needs to be corrected

Comments on the Quality of English Language

Minor corrections would be needed.

Author Response

Dear Reviewer,

Thank you very much for your insightful comment.

Ad.1 We were unable to obtain bigger population to our study. The COVID pandemia and organizational lacks (patient get the information about operation date far too late). The research was performed in one rehabilitation center and, for formal reasons it was not possible to add other outpatient clinic to the research. The authors agree, the bigger sample may give different results in statistical analysis.

Ad.2 Individual fitted therapy may give different results. In current literature however personalized training is not always the most effective. Both outpatient prehabilitation program as well as the leaflet were suited to the majority of most common biomechanical issues present in terminal stage arthrosis in patients’ population. The outpatient prehabilitation procedures were chosen considering the accessibility of procedures for any polish outpatient rehabilitation center. Similar procedures are impossible to perform at home conditions.

Ad. 3 I agree with the Editor. At the beginning simple 6 minute walk test (6MWT) was to be applied to estimate general walking efficacy (globally with circulatory and respiratory efficacy). Unfortunately, the outpatients’ rehabilitation building prevented us from performing the test. The pelvis stabilization was examined as well as global hip muscle strength. Obesity, occupational factors, genetic and other factors modify the course of disease. Probably some patients demand more complex interventions (diet modification, workplace and ergonomy interventions, lifestyle correction etc.) yet we were unable to provide it.

Ad.4 I have checked it. This is not a mistake.

Ad.5 It is mentioned in lines 268-276 of the version presented before revisions. The rehabilitation in  terminal stage coxartrosis cause more effort and needs patients’ determination. The hip arthroplasty is serious operation performed in the moment the symptoms of disease severely limits the everyday activity. The potential for improvement of general status without operation is poor. It is also possible that patients are not sufficiently motivated to rehabilitate before surgery, considering arthroplasty to be a sufficient intervention. Performing hip arthroplasty in earlier stage would probably improve prehabilitation effect.

On the other hand, the operations are performed late, as the predicted “prothesis lifespan” is ca.  20 years and prothesis replacement causes significantly larger risk of failure comparing to primary arthroplasty. I will try to present the arguments in more clear way.

Ad.6 The use of “prehabilitation” and “rehabilitation” in the text will be clarified.

Round 2

Reviewer 2 Report

Comments and Suggestions for Authors

Dear Authors,

Language editing was done resulting in a high quality manuscript. However, the influence of COVID-19 pandemic to this study was not adequately addressed. As you mentioned (lines 309-315) more than half of the participant were recruited after 2020. So, in order to be methodologically correct, two subgroups should be formed; participants recruited pre and during pandemic. Comparisons should be done to prove if these groups have similar characteristics or differ in a statistical significant way. Moreover, power analysis is important to be mentioned. It is not enough just a reference of small sample as study limitation (Line 327).  Last but not least, it is important to justify according to exist literature the limited prehabilitation to 15 days (in 3 weeks) - lines 154-155 and 171-174. What was the rationale for this exact period? In case of an extended period, results would be different? Regarding the kind, the duration and sets of exercise please provide data and reference proving that this was the most appropriate combination for ultimatum result.

Line 85: Please mention what stands for the abbreviation PMA.

Lines 149-153: Please provide some information regarding the rationale and measurement method of HOOS, VAS and PMA. What is mentioned in lines 151-153 is not acceptable in a scientific publication. You have to clarify the method using an acceptable reference.

Lines 181-183: Please provide the rationale supported by appropriate references of this delay between discharge and rehabilitation admission. Moreover, during this "dead" period, did participants perform any exercises or just lay upon beds?

Lines 199-200: please provide data for the meaning of this formula.

Line 211: As far as I am concerned table 5 is the correct  table regarding this paragraph.

Line 218: Please mentioned if the surgeon or the surgical team was the same in order bias to be reduced.  

Line 222: Revision arthroplasty is preferable to "realloplasty".

Lines 232-233: Please provide more information regarding the way in which "participants denied serious subjective improvement". Did you use any kind of scientifically accepted Patient Reported Outcome Measurement Score (PROMS)?

Lines 238-245 & 248-256 & 259-267: These three paragraphs are almost identical providing no useful information. Please rephrase. Moreover the last sentence of each paragraph is of no meaning.

Line 329: Please mention in a clear manner the organisational reasons why postoperative and rehabilitation planning is difficult to be made.

Author Response

Dear Editor,

Thank You for your reasonable and logical remarks. We are sure the manuscript will gain value thank to Your inquiry.

The pre COVID and COVID group characteristics has been investigated with no important differences revealed. The analysis is presented in the attachment. The results of our study will be useful to indicate parameters useful for further investigations and sample estimation.

The rehabilitation took 15 days due to National Health Fund regulations for outpatient rehabilitation and for organizational reason – the patients are not informed about planned alloplasty date with reasonable advance. The study design was to check the efficacy of prehabilitation in model possible to be commonly applied in Polish healthcare system. The Meta – Analysis by Gill and McBurney (reference  24 of modified version ) considered the data of prehabilitation studies with intervention duration ranging from 3 up to 12 weeks. In fact the patients were training 1 up to 5 times a week. Taking training 5 times a week means the intensity of intervention was bigger than usually used in other studies. The optimal intensity of exercise was not clarified before study started. There are multiple prehabilitation programs for hip arthrosis present in previous studies. However the “Hip alloplasty prehabilitation exercise golden standard program” was not established nor widely accepted.

Some interesting therapeutic options e.g. Tai Chi, aquatic exercise are not commonly available in Polish healthcare.  The additional references and explanations are added in the manuscript.

Abbreviation for PMA was explained (line 60 of modified version).

The line 149 -153 section was modified. The additional explanation was added (lines 131-138 and 170-177 of present manuscript version).

The text was changed. The preoperative and postoperative care standards have been established by  Health Technology Assessment and Tariffication Agency (in Polish Agencja Oceny Technologii Medycznych i Taryfikacji). The article is added to references. The first control visit after discharge with wound control and removing surgical suture should be conducted 12 -14 days after discharge from hospital. The regulations is shown with explanation of “dead period activity” mentioned in lines181-183 of previous version in lines 170-177 of present version of the manuscript.

The formula is clarified in lines 193-195. Thank You for remark, the original version was really misleading.

The table 5 is correct. Noone noticed it before. The number was corrected in the text.

Unfortunately we were unable to ensure the same surgical staff. It is mentioned in limitations section line 337-338. The majority of patients choose outpatient rehabilitation centers using distance criteria. Our rehabilitation center has peripheral location and we managed to admit as many patients as possible from different hospitals.

“Realloplasty” was corrected.

The PROM as term was described in 2009, but in fact there had been many self - reported patient tools recently. HOOS score and VAS are good examples of “PROMs before PROMs”. The opinion expressed by patients was a part of unofficial talks with patients during control visits. Sometimes patients say very valuable things, yet it is not a subject of the study. The authors agree that modern PROMs will be more commonly used in future. The PROMs were briefly accented in lines 134-137.

According to the recommendations of another Editor these paragraphs were added. The descritpions are pretty similar but describe different tables. The last sentence was removed.

The brief explanation of organizational disturbances is added in lines 327-329 and in lines 85 -89.

Reviewer 5 Report

Comments and Suggestions for Authors

Thank you for your answers.

Comments on the Quality of English Language

English language is fine.

Author Response

Dear Editor,

thank You for supporting development of our study and valuable remarks.